# OpenReview forum: "Safe RLHF-V: Safe Reinforcement Learning from Multi-modal Human Feedback"
_NeurIPS.cc/2025/Conference — NeurIPS 2025 poster_

### Official Review · Reviewer_NsRm · 2025-06-29

**Clarity:** 3
**Significance:** 3
**Originality:** 3
**Rating:** 5
**Confidence:** 3

**Summary:**

This paper proposes Safe RLHF-V, the first comprehensive framework for safe alignment of multimodal large language models (MLLMs). The method integrates three key components: (1) a dual-preference dataset (BeaverTails-V) with separate annotations for helpfulness and safety, including fine-grained severity labels; (2) a guardrail system (Beaver-Guard-V) that performs multi-level filtering through rejection and regeneration; and (3) a constrained reinforcement learning algorithm that optimizes for helpfulness under safety constraints using a min-max Lagrangian framework with a budget-bound strategy. Experimental results across five MLLMs and multiple benchmarks show consistent improvements in both safety (+34.2%) and helpfulness (+34.3%).

**Questions:**

See Weaknesses

**Ethical Concerns:**

["NO or VERY MINOR ethics concerns only"]

**Final Justification:**

As the authors have satisfactorily addressed my concerns, and given that my original score was positive, I am inclined to retain it.

**Quality:**

3

**Strengths And Weaknesses:**

Strengths:
This paper investigates the problem of aligning multimodal large language models (MLLMs) with both helpfulness and safety objectives. The authors propose a dual-objective alignment framework that separately models reward and cost, grounded in constrained optimization theory. A key contribution is the construction of the BeaverTails-V dataset, which includes diverse harmful content categories and high-quality dual annotations, offering a valuable benchmark for future research. In addition, the paper introduces Beaver-Guard-V, a multi-stage guardrail system that significantly improves robustness against adversarial queries via rejection and regeneration mechanisms. Extensive experiments and ablation studies across various models, evaluation metrics, and data scales comprehensively validate the effectiveness and stability of the proposed method.
Weaknesses
1. The generalization ability to other modalities (e.g., speech, video) is not evaluated. Although the current focus is on image-text, would the same framework apply in other multimodal contexts?
2. The subjectivity and consistency of the safety labels, particularly for minor/moderate/severe categories, is not fully discussed. Did the authors measure annotator agreement or validate against human gold standards?
3. While the Lagrangian min-max formulation is well-motivated, its comparison with other safety-constrained RL methods is missing?
4. How computationally expensive is Safe RLHF-V, especially the multi-round Beaver-Guard-V moderation? Can it be efficiently integrated into real-time inference?

---

> ### Author Rebuttal · Authors · 2025-07-31
>
> Dear Reviewer NsRm,
>
> Thank you very much for your insightful comments. We are encouraged by your positive comments regarding our work offering a valuable benchmark for future research and comprehensively validate the effectiveness and stability of the proposed method. We sincerely hope that **the following point-by-point responses** will adequately address your concerns and contribute to the acceptance of our paper.
>
> ---
>
> > **Q1. The generalization ability to other modalities (e.g., speech, video) is not evaluated. Although the current focus is on image-text, would the same framework apply in other multimodal contexts?**
>
> **R1:** We sincerely thank the reviewer for raising critical questions. As you pointed out, our paper primarily focuses on the image-text alignment task, around which we designed and validated the `Safe RLHF-V framework`. In particular, we constructed the BeaverTails-V preference dataset, where a central challenge lies in effectively collecting and distinguishing helpfulness and safety preferences under the visual modality, followed by constrained preference optimization based on dual signals.
>
> That said, as the reviewer insightfully noted, while the algorithmic framework is general, applying it to more complex modalities such as speech or video poses several engineering and system-level challenges:
> 1.	**`Complexity in preference data construction`:** Compared to image-text tasks, speech and video often carry stronger subjectivity. Especially for long-horizon or interactive content, defining and annotating “safety” and its levels remains an open challenge.
> 2.	**`Increased computational demands`:** Modalities like video entail significantly higher data dimensionality, leading to greater training complexity and computational requirements. This may also demand stronger stability and convergence guarantees in optimization.
> 3.	**`Finer-grained alignment requirements`:** Video and speech data typically involve more fine-grained information (e.g., frame-level or timestamp-level details), placing higher demands on the design of both the reward and cost models.
>
> We also hope the community will join us in exploring the feasibility of extending this constrained preference optimization framework to higher-dimensional modalities such as video-based dialogues or audio-based QA. In future work, we plan to further evaluate the Safe RLHF-V in these more complex settings and develop the corresponding datasets and training pipelines to advance the field of multimodal safety alignment
>
> ---
>
> > **Q2. The subjectivity and consistency of the safety labels, particularly for minor/moderate/severe categories, is not fully discussed. Did the authors measure annotator agreement or validate against human gold standards?**
>
> **R2:** We sincerely thank the reviewer for the insightful comment. As noted, Reviewer Dc2g raised a similar concern regarding how to ensure consistency between our current annotations and human judgments.
>
> **Safety is not a binary true-or-false attribute**, but rather a nuanced concept with varying degrees. In this work, we adopted a graded annotation scheme to capture the subtleties of safety levels. Due to cost constraints, GPT-4 was used as the primary annotator during the initial labeling phase. However, the entire dataset was subsequently validated by a professional annotation team with substantial experience in large model safety alignment.
>
> As shown in **Table 1**, we report the agreement rates between GPT-4 annotations and human annotators across different safety levels. Furthermore, **Table 2** presents the consistency analysis between human and AI assessments in evaluating model performance, providing additional evidence of annotation reliability.
> - We randomly sampled 2,000 preference annotation instances from outputs generated by various models.
> - All samples were independently evaluated by our long-term professional annotation team specializing in safety evaluation. This team has extensive experience in large model safety assessments and has participated in multiple relevant projects, ensuring both professional judgment and ethical awareness.
> - We then compared the human annotations with GPT-4o’s automatic evaluations to assess the alignment between the two.
>
> > **Note:** Due to our long-standing collaboration with this professional team, we were able to conduct this emergency annotation task within a short timeframe.
>
> **Table 1: The human and AI annotation agreement in different safety levels.**
> |                         | minor | moderate | severe |
> |-------------------------|-------|----------|--------|
> | Human and AI Agreement | 85.3% | 87.8%    | 89%    |
>
> **Table 2: The human and AI annotation agreement in total safety and helpfulness preference.**
> |                         | safety  | helpfulness  |
> |------|-------|----------|
> | Human and AI Agreement | 87.4% | 91.2%    |
>
> **With approval from the Institutional Review Board (IRB)**, the annotation of the BeaverTails-V dataset was carried out in collaboration with a professional annotation team. This team has extensive experience in large model safety alignment and has previously collaborated with us on several well-known alignment research projects. They demonstrate strong professional competence. For each batch of data, the accepted annotations maintained over 85% consistency with our spot-check evaluations, ensuring reliable annotation quality.
>
> ---
>
> > **Q3. While the Lagrangian min-max formulation is well-motivated, its comparison with other safety-constrained RL methods is missing?**
>
> **R3:** We sincerely thank the reviewer for the valuable suggestions. We have also conducted an in-depth investigation into safety-constrained RL methods.
>
> Notably, second-order methods like CPO [1] and PCPO [2] require computing the Hessian matrix of the model, which is computationally infeasible given the enormous parameter size of multimodal large models. Due to these resource constraints, we prioritized first-order approaches such as FOCOPS [3], and are currently in the process of replicating them under time constraints.
>
> **From a methodological perspective**, Safe RLHF-V validates the effectiveness of constrained preference optimization in multimodal safety alignment. Algorithms like FOCOPS are essentially extensions of constrained preference optimization under the CMDP framework, and are thus theoretically applicable and potentially effective in our setting as well.
>
> As this is **the first work to apply constrained preference learning to the multimodal safety alignment**, we overcame significant engineering challenges to demonstrate the viability of the framework. In our current experiments, we **adopted a standard PPO-Lagrangian for min-max optimization**, as it provides a simple yet effective baseline. `The experimental setup follows prior works such as Safe RLHF [4] and the original BeaverTails [5] paper.`
>
> We truly appreciate the reviewer’s suggestions. During the rebuttal period, we have already started preliminary integration attempts. However, incorporating more advanced safety-constrained RL methods into MMLMs requires a longer validation cycle. That said, we commit to including at least two additional safety-constrained RL baselines in the next version of our work to further enrich the comparison and strengthen our conclusions.
>
> [1] Constrained Policy Optimization
> [2] First Order Constrained Optimization in Policy Space
> [3] Projection-Based Constrained Policy Optimization
> [4] Safe RLHF: Safe Reinforcement Learning from Human Feedback
> [5] BeaverTails: Towards Improved Safety Alignment of LLM via a Human-Preference Dataset
>
> ---
>
> > **Q4. How computationally expensive is Safe RLHF-V, especially the multi-round Beaver-Guard-V moderation? Can it be efficiently integrated into real-time inference?**
>
> **R4:** We sincerely thank the reviewer for their valuable comments. The training cost of the Safe RLHF-V framework is indeed an important issue worthy of further investigation. In response, we conducted a detailed comparison between Safe RLHF-V and vision-based RLHF within the Verl framework. We used the Qwen2-VL-7B model and trained on the BeaverTails-V dataset (18k samples) using both PPO and Safe RLHF-V for 1 epoch. **The training was performed on 8*A100 GPUs.**
>
> Under the same experimental settings, PPO took 6 hours with a reduced memory footprint of 17,750 MB per GPU. Safe RLHF-V, trained under the same settings, took 7.57 hours, which is 26.1% slower than RLHF (see Table 2).
>
> **Table 2. Comparison of Safe RLHF-V and RLHF (in vision) Training Cost on Qwen2-VL-7B with the Beavertails-V (18k) Dataset**
> |Method|Time|#ofGPUs|Time-Compared-to-PPO-Default|Epochs|Batch-Size|Max-Response-Len|
> |---------------|-----|-------|----------------------------|------|----------|----------------|
> |PPO (in vision)|6.0h|8|~|1|4|512|
> |SafeRLHF-V|7.57h|8|+26.1%|1|4|512|
>
> [1] verl: Volcano Engine Reinforcement Learning for LLMs, https://github.com/volcengine/verl
>
> In current practice, companies offering online model services adopt highly cautious risk control strategies. **One effective approach is to integrate a Guard Model before or after the main model’s inference process**. This additional module serves to verify model inputs and outputs to enhance safety. The overall cost incurred depends primarily on the inference overhead of the Guard Model itself. In practice, models with 0.5B, 2B, or even 7B parameters can be used as Guard Models, and their cost is negligible compared to the main online model—for example, DeepSeek-R1 has 671B parameters.
>
> We sincerely thank the reviewer for their thorough and insightful feedback. We will incorporate the above analysis into the revised version of our paper and provide more quantitative experimental results on the inference cost of the Guard Model.

---

> > ### Comment · Reviewer_NsRm · 2025-08-05
> >
> > Thanks for the detailed response. While I still look forward to seeing future work that explores these directions more concretely, I believe the authors have adequately addressed the concerns I raised.
> >
> > Given that my initial review was already positive, I will maintain my current score.

---

> ### Author Response · Authors · 2025-08-05
> **Heartfelt Thanks for Your Thoughtful Review**
>
> Dear Reviewer NsRm,
> We sincerely thank you for the positive evaluation and thoughtful engagement throughout the review process. This has been a very meaningful rebuttal process for us, and we will incorporate the relevant content and improvements in the revised version of the paper. We look forward to advancing this line of work further.
>
> We are grateful for your support and constructive feedback, which have been invaluable in improving our work.
>
> Best regards,
> Authors of the paper

---

### Official Review · Reviewer_KXX1 · 2025-07-02

**Clarity:** 2
**Significance:** 3
**Originality:** 3
**Rating:** 4
**Confidence:** 3

**Summary:**

This article aims to investigate safety alignment fine-tuning techniques based on multimodal large models. Previous fine-tuning approaches based on Reinforcement Learning from Human Feedback (RLHF) have typically been applied to language-only large language models, with limited application to vision-language models, thus this paper partially addresses this gap. The contributions of this paper are threefold: first, it proposes a dataset for multimodal RLHF alignment; second, it designs a defense system against harmful outputs for multimodal large models; and third, it improves safe RLHF by dynamically updating the Lagrange coefficients in constrained optimization.

**Questions:**

Please see weaknesses. I will increase my rating if the authors can demonstrate the novelty of the RLHF-V algorithm.

**Ethical Concerns:**

["NO or VERY MINOR ethics concerns only"]

**Final Justification:**

I maintain my rating of 4 (Borderline Accept). The authors' rebuttal adequately addressed my concerns, but the core algorithmic contribution remains incremental - essentially adapting existing Safe RLHF to multimodal settings rather than introducing fundamental innovations.

Despite limited algorithmic novelty, the systematic contribution (comprehensive dataset, framework, and evaluation) addresses an important underexplored problem and provides solid value to the community.

**Limitations:**

It is recommended that the authors include a limitations section in the main text to provide a more detailed description of the study's limitations.

**Paper Formatting Concerns:**

No major formatting issues were identified in this paper.

**Quality:**

3

**Strengths And Weaknesses:**

**Strength**
1. The topic of multimodal alignment and the balance between its safety and usability has been underexplored and is of significant importance.

2. The workload of this paper is substantial, and its results are comprehensive. Centered on the theme of multimodal alignment, it not only proposes a new dataset (along with a data collection methodology) but also designs a harmful output detection system and introduces algorithmic innovations for safe RLHF. Overall, the content is systematic and robust.

**Weaknesses**
1. This work improves the algorithm based on safe RLHF, resulting in safe RLHF-V. However, the novelty appears limited. The authors dynamically update the Lagrange coefficient in Equations 7 and 8, increasing the coefficient to impose greater penalties when the cost is high, and decreasing it otherwise. For such constrained optimization problems, a common approach is to fix the value of λ (i.e., treating it as a hyperparameter for the penalty term, such as in L1-based weight sparsity). The contribution of this algorithm seems to lie solely in the dynamic adjustment of this coefficient.

2. There are some minor issues, such as the lack of explanation for the meanings of the variables in Equations 7 and 8. It is recommended to review the readability of the entire text and include additional details in the main body.

---

> ### Author Rebuttal · Authors · 2025-07-31
>
> Dear Reviewer KXX1:
>
> We sincerely thank you for the positive evaluation and recognition of our work. We are grateful that you acknowledged the research problem we address as underexplored and of significant importance, and that you found the workload of this paper to be substantial, with systematic and robust content. We also deeply appreciate your careful review and the constructive suggestions provided.
>
> Below, we respond to each of your comments in detail. We hope our responses can address your concerns thoroughly.
>
>
> > **Q1(1): This work improves the algorithm based on safe RLHF, resulting in safe RLHF-V. However, the novelty appears limited.**
>
> **R1(1):** We sincerely thank the reviewer for both the recognition and the thoughtful critiques of our work. As you pointed out, Safe RLHF-V builds upon existing Safe RLHF methods. However, we would like to emphasize that our contribution goes beyond a straightforward extension from the textual modality to the multimodal setting. Instead, we design a comprehensive pipeline tailored to the challenges of multimodal safety alignment, which includes **a dataset**, **an alignment algorithm**, and **a defense system**. **`As you noted in the strengths section, this constitutes a substantial workload, which we believe is a key contribution of this paper.`**
>
> Specifically, our work includes the following additional contributions beyond algorithmic improvements:
> 1.	We release BeaverTails-V, the first large-scale multimodal preference alignment dataset covering 9 primary and 20 secondary safety categories (as shown in Figure 5). While GPT-4o was primarily used for annotation, we additionally conducted a human-AI consistency evaluation, showing over 85% agreement between human experts and GPT-4o, as reported in **Tables 1 and 2**. Moreover, we introduce a hierarchical safety severity framework (Safe / Minor / Moderate / Severe), which is, to the best of our knowledge, the first of its kind in multimodal safety preference alignment.
> 2.	Based on the dataset, we propose BeaverGuard-V, a training-free defense system that filters unsafe outputs without requiring any model fine-tuning. It performs multi-level safety judgment at both the input and output stages, and reduces the proportion of unsafe content by 50.2%, all without modifying the base model parameters. This plug-and-play mechanism offers strong practical value for real-world deployment, especially in risk control scenarios for companies providing public access to large models.
> 3.	Finally, we develop Safe RLHF-V, a constrained preference optimization algorithm for multimodal alignment. Given the inherent instability of multimodal training and the convergence challenges posed by tuning the Lagrangian multiplier λ in constrained optimization, we introduce a Min-Max Optimization with Budget Bound strategy to improve both convergence stability and safety performance. To the best of our knowledge, this is the first successful extension of Safe RLHF to the multimodal alignment setting.
>
> **Table 1: The human and AI annotation agreement in different safety levels.**
> |                         | minor | moderate | severe |
> |-------------------------|-------|----------|--------|
> | Human and AI Agreement | 85.3% | 87.8%    | 89%    |
>
> **Table 2: The human and AI annotation agreement in total safety and helpfulness preference.**
> |                         | safety  | helpfulness  |
> |-------------------------|-------|----------|
> | Human and AI Agreement | 87.4% | 91.2%    |
>
>
> In summary, we believe that the contribution of our work goes far beyond algorithmic improvements. To the best of our knowledge, this is the first systematic study of multimodal safety alignment, covering the entire pipeline from data construction, algorithmic design, to training and deployment-time evaluation.
>
> We once again sincerely thank the reviewer for the generous evaluation and constructive feedback. Your thoughtful comments have motivated us to clarify the contributions of our work more thoroughly. We hope the above clarifications can help address your concerns effectively.
>
> ---
>
> > **Q1(2): The authors dynamically update the Lagrange coefficient in Equations 7 and 8. For such constrained optimization problems, a common approach is to fix the value of λ (i.e., treating it as a hyperparameter for the penalty term, such as in L1-based weight sparsity).**
>
> **R1(2):** We sincerely thank the reviewer for your attention to the details of the constrained optimization. You are absolutely correct in understanding that Equations (7) and (8) are responsible for this functionality.
>
> While fixing the Lagrange multiplier λ, as commonly done in L1 regularization, offers theoretical simplicity and controllability, we found that this approach encounters significant practical challenges in reinforcement learning, especially in multimodal RLHF settings:
> - High variability in the reward and cost spaces: In multimodal alignment, the distribution of rewards and costs fluctuates significantly across different model architectures and input modalities. A fixed λ value cannot adequately adapt to the dynamic shifts in reward/cost distributions during different training phases.
> - Training instability and convergence failure: Fixed λ values often lead to training collapse or ineffective safety alignment. In our experiments, when λ is set too high, the model receives excessive penalties, causing both rewards and costs to plummet, leading to instability or divergence. Conversely, if λ is too small, the model fails to enforce meaningful safety alignment, resulting in limited improvement in safety.
>
> To empirically validate this point, we conducted comparative experiments between dynamic λ scheduling and fixed λ values using LLaVA-1.6-7B. The results are presented below:
>
> **Table 3: Dynamic vs Fixed Lagrange Coefficient in Safe RLHF-V.**
> |                        | Safety win rate | Helpful win rate |
> |------|-----------------|--------|
> | dynamic: λ = 10       | 0.58          | 0.75           |
> | dynamic: λ = 0.1      | 0.53          | 0.77           |
> | fixed: λ = 10         | 0.41          | 0.33           |
> | fixed: λ = 0.1        | 0.40          | 0.75           |
>
> As shown in the Table 3, dynamic λ values strike a better balance between safety and helpfulness, and maintain strong robustness across different initialization conditions. We also provide a detailed analysis in Section 4.3: Ablation Study of the paper. In contrast, fixed λ settings are highly sensitive and prone to instability depending on the specific value chosen.
>
> Therefore, we argue that the dynamic λ update mechanism is not merely a technical detail, but rather a key enabler of stable convergence in complex multimodal RLHF settings. Although similar techniques have proven effective in pure-text scenarios, our work makes the first exploration of safe alignment in Multimodal LLMs (MMLMs), and we believe this constitutes a scientifically significant contribution to the field.
>
> ---
>
> > **Q2：There are some minor issues, such as the lack of explanation for the meanings of the variables in Equations 7 and 8. It is recommended to review the readability of the entire text and include additional details in the main body.**
>
> **R2:** Thank you very much for pointing out this important readability issue. We sincerely apologize for the confusion caused by the lack of detailed explanations for the key variables in the formulas. Upon receiving your comments, we immediately organized a thorough review of the manuscript among the author team.
>
> Based on the feedback collected, we have revised the manuscript to explicitly define and explain all variables and mathematical symbols used in Equations (7) and (8). In addition, we have standardized other potentially ambiguous notations and annotations throughout the paper.
>
> We are committed to further refining the manuscript in the final version, with particular attention to improving the clarity of technical details and mathematical formulations. We sincerely thank the reviewer for the valuable suggestions, which have significantly helped us enhance the clarity and readability of our work.
>
> `If our point-by-point responses have sufficiently addressed your concerns, we would sincerely appreciate your kind consideration of a more higher score to support the acceptance of our paper`. **We truly value the time and effort you have dedicated during the review process, and we are deeply grateful for your constructive feedback.**

---

> > ### Comment · Reviewer_KXX1 · 2025-08-05
> >
> > I thank the authors for their thorough responses during the rebuttal period. Your detailed explanations have addressed most of my initial concerns.
> >
> > While the core algorithmic contribution remains somewhat incremental as I initially noted, the comprehensive nature of the framework and its potential impact on the multimodal safety alignment field are valuable. The experimental validation is thorough and the results are convincing.

---

> > > ### Author Response · Authors · 2025-08-05
> > > **Heartfelt Thanks for Your Thoughtful Review**
> > >
> > > Dear Reviewer KXX1,
> > >
> > > We sincerely thank you for your recognition and valuable feedback. This has been a truly meaningful review process for us. We are pleased that the additional experimental results and explanations have been able to resolve your concerns.
> > >
> > > We are also deeply grateful for your positive view of the overall framework and its contributions to the field of multimodal safety alignment. We truly appreciate your recognition, and we are committed to incorporating the content discussed during the rebuttal into the revised version.
> > >
> > > Best Regards,
> > > Authors of the paper

---

### Official Review · Reviewer_yjaW · 2025-07-02

**Clarity:** 3
**Significance:** 2
**Originality:** 2
**Rating:** 3
**Confidence:** 4

**Summary:**

This paper introduces Safe RLHF-V, the first comprehensive framework for safety alignment in multimodal large language models (MLLMs), which integrate visual and textual inputs. Traditional RLHF methods often optimize for helpfulness alone, neglecting safety, especially in the multimodal setting. To address this, the authors propose a dual-objective alignment approach that simultaneously optimizes for helpfulness and safety via constrained reinforcement learning.

**Questions:**

please see weakness

**Ethical Concerns:**

["Major Concern: Data privacy, copyright, and consent"]

**Final Justification:**

This paper focuses primarily on dataset construction and presents limited methodological novelty. I have maintained a negative rating due to the lack of real human involvement in the dataset creation process, which may introduce bias and limit the dataset's reliability.

**Quality:**

3

**Strengths And Weaknesses:**

**Strengths**

1. Safe RLHF-V is the first to explicitly disentangle and optimize both helpfulness and safety in multimodal alignment.

2. The authors construct BeaverTails-V, a well-annotated dataset that better reflects harmful multimodal interactions, and Beaver-Guard-V, a robust moderation system with fine-grained QA-pair severity filtering, improving both evaluation and real-world safety.

3. The paper is well-written and easy to follow.

**Weaknesses**

1. The novelty is limited. While I acknowledge the effort to disentangle and optimize both helpfulness and safety in multimodal alignment, I do not consider this a strong novelty on its own.

2. The authors claim that constrained optimization has shown effectiveness in language models, but that training becomes highly unstable in multimodal settings. This point is not well-justified. First, the issue that reward shaping tends to improve only a single dimension is not unique to the multimodal setting—it also exists in other domains. Thus, the motivation is not very convincing. Second, as shown in Figure 3, some static reward shaping settings actually lead to improvements in both metrics, which weakens the authors' claim.

3. Much of the data, including prompts, responses, and annotations, is generated or evaluated by LLMs like GPT-4o, which may inherit biases or inconsistencies. The lack of human-annotated validation raises questions about reliability.

---

> ### Author Rebuttal · Authors · 2025-07-31
>
> Dear Reviwer yjaW,
>
> **Thank you very much for your thorough review and the time you devoted to evaluating our work.** We have carefully read your comments and provided **a detailed point-by-point response** addressing each of your concerns. In particular, we **clarified that the core contribution of our paper goes beyond a single algorithm** — it includes a comprehensive dataset, a practical solution, and a full defense system pipeline. We also **provided a clear explanation of the fundamental differences between Safe RLHF-V and reward shaping**, and **conducted an important analysis of human-AI alignment consistency**.
>
> **`We sincerely hope that our responses address your concerns and demonstrate the value of our work. We would be truly grateful if you recognize the positive impact our contribution may have in advancing research on multimodal safety alignment.`**
>
>
> > **Q1. The novelty is limited.**
>
> **R1:** We thank the reviewer for raising critical questions about the contributions of our work. As you pointed out, Safe RLHF-V is an extension of the existing Safe RLHF framework. However, we would like to clarify that **our work is not a straightforward application of Safe RLHF from the text modality to the multimodal setting**. Rather, it builds a comprehensive pipeline for multimodal safety alignment, encompassing a dataset, algorithm, and defense system — as you also recognized in the Strengths section, which we believe represents **a substantial workload for a single paper**.
>
> In constructing this pipeline, our contributions go beyond algorithmic adaptations. Specifically, the paper also introduces the following additional components:
> 1. We release BeaverTails-V, the first large-scale multimodal preference dataset with 9 primary and 20 secondary safety categories, featuring a novel four-level hierarchical safety annotation.
> 2. We introduce BeaverGuard-V, a training-free defense system that performs multi-level safety filtering at both input and output stages, reducing unsafe content by 50.2% without model fine-tuning.
> 3. We propose Safe RLHF-V, the first constrained preference optimization method successfully applied to multimodal alignment.
>
> Reviewer Dc2g commented that our work:
> > **One of the major highlights of this study** is ... open-sourcing of the BeaverTails-V dataset. ... This design addresses existing shortcomings in current datasets, thus **offering a valuable resource for future research**.
>
> Not limited to the dataset, the Safe RLHF-V framework primarily provides a more comprehensive solution for ...
> > **Rather than being limited to a single algorithm, the authors provide a comprehensive end-to-end framework—including dataset construction, model defense, and alignment optimization**
>
> Reviewer yjaW also comment our paper:
> > Safe RLHF-V is the first to explicitly disentangle and optimize both helpfulness and safety in multimodal alignment.
>
> Reviewer KXX1:
> > **The topic of multimodal alignment** and the balance between its **safety** and usability **has been underexplored** and is of significant importance.
> > **The workload of this paper is substantial**, and its results are comprehensive. ... it not only proposes a new dataset (along with a data collection methodology) but also designs a harmful output detection system and introduces algorithmic innovations for safe RLHF. Overall, the content is systematic and robust.
>
> **We fully understand the authors’ concerns, especially regarding the algorithmic aspects. However, we sincerely hope that the reviewers will also recognize the overall contribution of our work—particularly its positive impact in encouraging further exploration of multimodal safety within the community, and consider adjusting the score accordingly to give us a chance for acceptance.**
>
>
> > **Q2. The authors argue that constrained optimization is unstable in multimodal settings, but this is not well-supported. Reward shaping’s limitation to a single dimension is not unique to multimodality, weakening the motivation. Moreover, Figure 3 shows that some static shaping methods improve both metrics, further challenging their claim.**
>
> **R2:** Thank you very much for your valuable feedback. We provide a quantitative comparison analysis of the cost model between Vicuna-7B (text-only) and LLaVA-1.5-7B (multimodal), both trained under exactly the same conditions using the Safe RLHF pipeline, without introducing the scaling hyperparameter k proposed in this work for multimodal settings.
>
> As shown in the Table 1, cost model in the multimodal setting deteriorates compared to the text-only:
> - 1) across multiple distributional metrics, the separation between safe and unsafe responses clearly decreases: the mean difference between unsafe and safe responses drops from 5.069 to 2.237; ROC-AUC decreases from 0.958 to 0.888; the overlap coefficient increases from 0.211 to 0.371.
> - 2) the decision boundary between safe and unsafe distributions shifts away from zero: the optimal classification threshold moves dramatically from −0.054 to +0.656. These consistent degradation patterns indicate that multimodal inputs lead to severe compression and shift in the cost distribution.
>
> Therefore, due to the distribution compression and decision boundary shift, the traditional cost model and Safe RLHF training framework designed for text-only models can no longer effectively address this instability. To this end, 1) we introduce the scaling hyperparameter k to mitigate optimization difficulties caused by reduced separability; 2) we introduce the budget bound mechanism to adapt to the non-zero optimal classification threshold. Together, they enable stable and reliable constrained optimization in multimodal scenarios.
>
> **Table1.Performance Degradation of the Cost Model in Multimodal vs. Text-Only Settings.**
> |Metric|**Text-Only(Vicuna-7B)**|**Multimodal(LLaVA-1.5-7B)**|
> |----|----|----|
> |**ΔMean(Unsafe-Safe)$\uparrow$**|**5.069**|2.237|
> |**ROC-AUC$\uparrow$**|**0.958**|0.888|
> |**Overlap Coefficient$\downarrow$**|**0.211**|0.371|
> |**KSS tatistic$\uparrow$**|**0.788**|0.623|
> |**Jensen-ShannonDivergence$\uparrow$**|**0.661**|0.433|
> |**Wasserste in Distance$\uparrow$**|**5.069**|2.237|
> |**Optimal Classification Threshold [1]**|**-0.054**|+0.656|
>
> [1] Youden, W. J. (1950). Index for rating diagnostic tests. Cancer, 3(1), 32-35.
>
>
> **We provide a detailed explanation of Figure 3. $\downarrow$**
>
> On the left side of Figure 3, we present experiments on static reward shaping with varying initial values of $\lambda$. As shown, in most cases (e.g., $\lambda = 0.01, 0.1, 1$, represented by the blue dots), the model achieves improvements in safety performance, since $\lambda$ leverages the cost signal to apply a safety penalty during training. However, when the initial value of $\lambda$ is too large (e.g., $\lambda = 10$ or $100$), reward shaping fails to effectively enhance the model’s performance while ensuring safety—these two blue dots even fall below the performance of the unaligned model.
>
> In contrast, Safe RLHF-V, consistently achieves simultaneous improvements in both helpfulness and safety across all initializations ($\lambda = 0.01, 0.1, 1, 10, 100$). The red dots are notably concentrated around the diagonal line **$y = x$**, indicating that our approach is **not only robust to the initialization of $\lambda$**, **but also more balanced compared to static reward shaping—improving both dimensions of performance concurrently**. For example, consider the case with $\lambda = 0.01$ (blue dot). Due to the low initial value, reward shaping cannot dynamically adjust the penalty strength, leading to inefficient training: although helpfulness improves, safety gains are minimal.
>
> ---
>
> > **Q3: Much of the data, including prompts, responses, and annotations, is generated or evaluated by LLMs like GPT-4o, which may inherit biases or inconsistencies. The lack of human-annotated validation raises questions about reliability.**
>
> **R3:** We sincerely thank the reviewer for the insightful comment. As noted, Reviewer `Dc2g` and `KXX1` also raised a similar concern regarding how to ensure consistency between our current annotations and human judgments.
>
> **Safety is not a binary true-or-false attribute**, but rather a nuanced concept with varying degrees. In this work, we adopted a graded annotation scheme to capture the subtleties of safety levels. Due to cost constraints, GPT-4 was used as the primary annotator during the initial labeling phase. However, the entire dataset was subsequently validated by a professional annotation team with substantial experience in large model safety alignment.
>
> - We randomly sampled 2,000 preference annotation instances from outputs generated by various models.
> - All samples were independently evaluated by our long-term professional annotation team specializing in safety evaluation.
> - We then compared the human annotations with GPT-4o’s automatic evaluations to assess the alignment between the two.
>
> > **Note:** Due to our long-standing collaboration with this professional team, we were able to conduct this emergency annotation task within a short timeframe.
>
> **Table 1: The human and AI annotation agreement in different safety levels.**
> |                         | minor | moderate | severe |
> |---|--|----|--|
> | Human and AI Agreement | 85.3% | 87.8%    | 89%    |
> **Table 2: The human and AI annotation agreement in total safety and helpfulness preference.**
> |                         | safety  | helpfulness  |
> |--|--|---|
> | Human and AI Agreement | 87.4% | 91.2%    |
>
> **With approval from the Institutional Review Board (IRB)**, the annotation of the BeaverTails-V dataset was carried out in collaboration with a professional annotation team. This team has extensive experience in LLMs safety alignment. They demonstrate strong professional competence. For each batch of data, the accepted annotations maintained over 85% consistency with our spot-check evaluations, ensuring reliable annotation quality.

---

> > ### Comment · Reviewer_yjaW · 2025-08-02
> >
> > Thanks for your response.
> >
> > 1. I noticed that the overall agreement between human and AI annotators is below 90%. I am concerned that this level of discrepancy, over 10%, may raise questions about the reliability of the dataset. From my perspective, this suggests that models trained or evaluated on this dataset might be more aligned with GPT-style outputs rather than human judgment. Additionally, based on the current version of the paper, it appears that no human annotators were involved in the original labeling process. Therefore, I believe the claim of “substantial effort" unconvincing.
> >
> > 2. Since the primary contribution of this work lies in dataset construction, and the methodological components do not appear to be particularly novel, this paper may be a better fit for the Dataset and Benchmark track rather than the main track?

---

> ### Author Response · Authors · 2025-08-03
> **Response from Authors (Round 2)**
>
> Dear Reviewer yjaW,
>
> Thank you very much for your prompt response.
> > We sincerely apologize for the delayed reply. The paper's author has been experiencing a high fever, which worsened last night. We kindly ask for your understanding and appreciate your patience.
>
>  Please find our detailed response below.
>
> **Response (Q1: (Human-AI Agreement & Dataset Reliability)):**
>
> We appreciate the reviewer’s concerns regarding the dataset’s reliability. It is important to clarify that achieving over 95% or even 100% agreement between human and AI annotators is extremely rare in real-world tasks due to the inherent subjectivity and diversity of human preferences. **Even among professional human annotators, agreement typically maxes out around 70%–85%.**
>
> For example, **the DPO paper [1] (Section 6.4, Table 2) shows that GPT-4’s preferences align with human annotations `at only 65–87%`**, even for relatively simple TL;DR summarization tasks, depending on prompt batches. Similarly, OpenAI’s RLHF paper [2] reports:
> > Inter-annotator agreement among trained labelers is 72.6 ± 1.5%, and for held-out labelers, it is 77.3 ± 1.3%.
>
> In our rebuttal process, we additionally conducted a human-AI agreement analysis. The results show that human preferences align with the AI-labeled preferences at 91.6% for Helpfulness and 87.4% for Safety, which we believe is a strong indicator of the dataset’s quality and reliability in complex real-world settings.
>
> **Regarding the concern that “no human annotators were involved,”** we indeed supplemented this part during the rebuttal phase in response to suggestions from reviewers `Dc2g` and `KXX1`. With the help of our long-term professional annotation team, and with institutional IRB approval, we conducted a human evaluation experiment to validate AI annotations. We believe this reflects a sincere and substantial effort on our part to improve the work and address the reviewer’s concern.
>
> Moreover, **using API-based Models like GPT-4 as preference judges has become a widely accepted and efficient evaluation and annotation approach in many recent works** (also considering the constraints of time and cost.):
> 1. In OpenAI’s GPT-4 Technical Report (https://openai.com/index/gpt-4-research), AI evaluation was used to assess performance improvements, and an open-source framework for automated evaluation of AI model performance was released (https://github.com/openai/evals).
> 2. The LLaVA series of works has been widely recognized in the multimodal domain and frequently adopts this evaluation protocol. Successive versions such as LLaVA-1.5 and LLaVA-NeXT have retained this AI evaluation process.
> 3. MiniGPT-4: Enhancing Vision-Language Understanding with Advanced Large Language Models also follows this evaluation paradigm.
> 4. UltraFeedback: Boosting Language Models with Scaled AI Feedback
>
> [1] Direct Preference Optimization: Your Language Model is Secretly a Reward Model
> [2] Training language models to follow instructions with human feedback
>
> **Response (Q1: (Track Suitability and Primary Contribution)):**
> We understand the reviewer’s suggestion that the paper may be better suited for the Dataset and Benchmark track. However, we would like to clarify that the dataset is a supportive component, not the primary contribution of our work.
>
> > To the best of our knowledge, this work is the first successful adaptation of constrained preference optimization to multimodal safety alignment.
>
> Our core objective is to propose an effective approach for safe alignment of multimodal large language models. From the perspective of constrained preference alignment, we introduced the Safe RLHF-V algorithm. During validation, due to the lack of suitable decoupled preference datasets in the community—especially for the text modality—we additionally released the BeaverTails dataset and designed the Beaver-Guard-V system.
>
> Therefore, while the dataset itself is a valuable contribution, the methodological design and the alignment framework also represent the core technical innovations of our work. We believe our submission aligns well with the Main Track’s expectations in terms of both novelty and potential impact on the community.
>
> As authors—and also as reviewers for both the Main Track and the DB Track—we fully acknowledge the reviewer’s concerns. We genuinely hope that through clear and constructive communication, we can address these concerns and clarify the value of our contributions. We are optimistic that any misunderstandings can be further resolved.
>
> ---
>
> We sincerely welcome additional feedback from the reviewer. We appreciate the reviewer’s time and thoughtful comments, and we hope the clarifications provided will facilitate a positive reassessment of our submission.

---

> ### Author Response · Authors · 2025-08-05
> **Hoping to Resolve the Misunderstanding - We Sincerely Hope for Understanding and Respect**
>
> Thank you for your understanding. We noticed your previous comment was deleted, but no response was given to our message.
>
> > `However, since your previous comment has already been circulated via email, we kindly request that you provide a clarifying response to your earlier comment to avoid influencing the evaluations of other reviewers.`
>
> We respectfully ask that you reevaluate our work based on the substance of our rebuttal.
>
> During the rebuttal period, we engaged in two rounds of meaningful communication and invested considerable effort in addressing your concerns. Our responses included additional investigations, new experiments, and extended analysis—all grounded in academic rigor and ethics.
>
> Regarding Reviewer `Dc2g`’s comment, we were entirely unaware of it and responded sincerely to your reviews. We remain committed to constructive dialogue and hope your evaluation reflects the scientific merit of our work, not external or non-academic factors.
>
> We have further **responded to Reviewer Dc2g’s comment** and have also expressed our hope that they would consider **removing the defense**. We genuinely hope this misunderstanding can be resolved.
>
> > Although I initially considered raising my rating, in light of these issues, I believe it is more appropriate to leave it unchanged and suggest that the Area Chairs review the integrity of the review process for this submission.
>
> As authors, we cannot influence other reviewers’ actions and are uncertain whether Reviewer Dc2g saw our earlier response. Nonetheless, we did our utmost to address all raised issues. Safe RLHF-V is fully open-sourced, with public data and code aligned with NeurIPS policies. We hope our transparency and diligence help restore your confidence and resolve any remaining concerns.

---

> ### Comment · Reviewer_Dc2g · 2025-08-05
> **Response to authors**
>
> I am truly sorry that my inappropriate actions have led to a misunderstanding between you and the reviewer. However, I also feel that your wording may have been overly strong. To help de-escalate, I’ve asked the reviewer to consider removing their comment, and I wonder if you might also be willing to delete yours. I sincerely apologize that my actions have caused this misunderstanding.

---

> ### Comment · Reviewer_yjaW · 2025-08-05
>
> Dear Author and Reviewer Dc2g,
>
> Thank you for your clarifications.
>
> 1. First of all, regarding the quality of the paper, I am not opposed to its acceptance.
>
> 2. I have never questioned the integrity of Reviewers KXX1 and NsRm, who also gave positive ratings. I believe their reviews are insightful and can genuinely help improve the manuscript.
>
> 3. According to the policy, if the discussion phase is permitted, I believe I have the right to raise concerns about why a reviewer gave the highest score with the highest confidence. It should not be the case that only reviewers with negative ratings are subject to scrutiny.
>
> 4. What struck me as most unusual is that Reviewer Dc2g was able to run the experiments and respond within such a short window (I raised the issue on August 2nd and received a response by August 4th). The speed and nature of this defense seemed unusually proactive and raised questions for me.
>
> In short, if the area chair has no concerns about the review process, I am not opposed to the paper being accepted. Good luck!

---

> > ### Comment · Reviewer_Dc2g · 2025-08-06
> > **Response to reviewer yjaW**
> >
> > **a major misunderstanding!**
> > **a major misunderstanding!**
> >
> > There appears to be **a major misunderstanding**. I did not use the author’s dataset and code within two days, as has been implied.
> >
> > As clearly indicated by the authors in the checklist (submitted manuscript), line 1130 to line 1136:
> >
> > > 5. Open access to data and code
> > > Question: Does the paper provide open access to the data and code, with sufficient instruc-1131tions to faithfully reproduce the main experimental results, as described in supplemental material?
> > > Answer: [Yes]
> > > Justification: **Our of our data and code can be found at https://github.com/SafeRLHF-V**
> >
> > **The authors had already made both the dataset and the code publicly available in the original manuscript (line 1130 to line 1136).** During the review period, I did not have the time to re-run any of their experiments—let alone within two days. Given that the materials were already open-sourced, any suggestion otherwise is a misunderstanding.
> >
> > Coincidentally, we have recently been conducting related experiments in this area, exploring multimodal models including vision-language models and video models. Therefore, during the review process, we did run some simple experiments using the authors’ publicly available dataset and code. The results we obtained were quite promising.
> >
> > I have already asked the authors to refrain from making excessive or unwarranted accusations, and I hope this helps resolve the misunderstanding.
> >
> > This matter has been troubling me for the past two days. I sincerely apologize to you for any confusion it may have caused.

---

> > ### Author Response · Authors · 2025-08-06
> > **Official Comment by Authors**
> >
> > Dear Reviewer yjaW and Dc2g,
> >
> > We have removed the previously inappropriate comment and sincerely apologize for the overly strong language used earlier to Reviewer yjaW. This is our sincere apology for our previous excessive behavior.
> >
> > We genuinely appreciate the time and effort both you have dedicated to the review process. However, we remain uncertain about the source of the misunderstanding. Our paper includes fully open-sourced data and code, with an anonymized access link provided directly in the submission.
> >
> > We want to reaffirm that we have strictly adhered to all relevant submission guidelines throughout the entire process. To us, academic integrity is of the utmost importance. At the same time, we respectfully urge Reviewer yjaW to recognize the nature of this misunderstanding. We have devoted nearly two weeks to carefully preparing this rebuttal; one of our teammates even fell ill due to overwork during this period, which reflects the seriousness and sincerity of our response. We are concerned that the misunderstanding could lead to an unfair evaluation of our work.
> >
> > Given that the open-source code is verifiable and was made available in the paper, we sincerely hope that Reviewer yjaW will consider offering an apology for the misunderstanding caused.

---

> > > ### Comment · Reviewer_yjaW · 2025-08-07
> > >
> > > Dear Author,
> > >
> > > 1. I would like to update my score to reflect the quality of the paper.
> > >
> > > 2. I want to clarify that I have never questioned the integrity of the authors. My previous comments were solely in reference to the actions taken by Reviewer Dc2g.
> > >
> > > 3. I am sorry for any inconvenience may have caused.

---

> ### Author Response · Authors · 2025-08-07
> **Heartfelt thanks for your dedication and the responsibility, from the bottom of our hearts.**
>
> Dear Reviewer yjaW,
>
> Thank you very much for your understanding. This is the first time we have encountered such a situation, and we recognize that this has been a rather unique review process. In the meantime, we sincerely appreciate your dedication to the integrity of the review process and the considerable time and effort you have invested in engaging with the discussion.
>
> Please also trust that to us, academic research is sacred. In our hearts, the purity and integrity of science are more important than anything else.
>
> Once again, we sincerely apologize for our earlier actions, and we are truly grateful for your understanding.

---

### Official Review · Reviewer_Dc2g · 2025-07-06

**Clarity:** 3
**Significance:** 4
**Originality:** 4
**Rating:** 6
**Confidence:** 5

**Summary:**

This paper presents a framework named Safe RLHF-V, designed for safety alignment of MLLMs. The authors tackle the challenge by balancing the often conflicting objectives of helpfulness and safety within a constrained optimization framework, which is a meaningful topic in safety alignment.

The main contributions of this work are threefold: (1) The authors created and open-sourced the BeaverTails-V dataset, which is the first multimodal dataset featuring both helpfulness and safety preference annotations, with multi-level safety labels in LLM; (2) The authors developed Beaver-Guard-V framework, which is a multi-level guardrail system for proactively defending against unsafe queries and attacks; (3) The authors proposed Safe RLHF-V, which is the first alignment algorithm for multimodal settings that balances safety and helpfulness based on constrained optimization. Experimental results show that this framework significantly enhances MLLM's safety while preserving its helpfulness.

**Questions:**

1. Constructing the BeaverTails-V dataset is undoubtedly a major contribution, but since it contains diverse categories of harmful content, what specific ethical review measures were taken during data collection and annotation to protect annotators’ physical and mental well-being, and to prevent misuse of the data?
2. The paper convincingly demonstrates that decoupling “helpfulness” and “safety” preferences leads to superior alignment. Beyond the technical improvements, what key insights have the authors gained from this dual-objective approach? Might this paradigm of decoupling competing values be generalized to other alignment challenges, such as honesty or fairness in AI?
3. Beaver-Guard-V performs strongly across multiple benchmarks. However, how well does the model generalize? For subtle or novel attack types beyond the 20 categories covered in the BeaverTails-V dataset, does the model still provide robust defenses?

**Ethical Concerns:**

["NO or VERY MINOR ethics concerns only"]

**Final Justification:**

This article provides a valuable reinforcement learning alignment method that effectively balances "helpfulness" and "safety". Additionally, the authors fully open-sources the dataset and code,  which holds significant implementation value for both academia and industry. The paper is well-written and the experiments are thorough and well-documented. For each reviewer's comment, the authors have provided thorough explanations during the rebuttal stage. I believe this article is highly valuable and deserves to be accepted by this conference. Final rating: 6

**Limitations:**

yes

**Quality:**

4

**Strengths And Weaknesses:**

Strengths:

This paper is among the first comprehensive studies to explore safety alignment of MLLM. As the capabilities of MLLMs rapidly increase, their associated safety risks become ever more important. This work directly confronts this key challenge, with the proposed Safe RLHF-V framework establishing an important benchmark for the entire field. I found there are some other framework in LLM domain like PKU Safe-RLHF, but for MLLM, this work is the first.

1. One of the major contribution of this work is the creation of the BeaverTails-V dataset, which innovatively introduces independent preference annotations for helpfulness and safety, and provides multi-level safety labels (“mild,” “moderate,” and “severe”). This design addresses existing shortcomings in current datasets, such as the coupling of preference signals and insufficient safety-related annotations, thus offering a valuable resource for future research.

2. Rather than being limited to a single algorithm, the authors provide a comprehensive end-to-end framework—including dataset construction, model defense, and alignment optimization, which collectively comprise a systematic solution and greatly enhance the practical impact of this work.

3. The work models the safety alignment problem as a min-max optimization task based on the Lagrangian method, providing solid theoretical foundations. To address the instability in multimodal training, the authors introduce the concept of Budget Bound. The experimental design is thorough, validating the effectiveness of the proposed approach across multiple mainstream MLLMs, with convincing results.

4. The authors commit to open-sourcing all datasets, models, and code, an approach that is highly commendable. This offers great implementation value for both academia and industrial applications.

Weaknesses:

1. The authors correctly identify instability in multimodal alignment training as a critical challenge and propose the “Budget Bound” as a mitigation. However, the paper lacks a thorough analysis of the fundamental causes of such instability. Is it solely attributable to the updates of Lagrange multipliers? Or could it also stem from the visual encoder, methods of vision-language feature fusion, or inherent noise in multimodal data?

2. While PPO is a classic algorithm in RLHF, its training complexity, high computational cost, and sample inefficiency are well known. Compared to lighter-weight alignment methods like DPO, does the performance improvement of Safe RLHF-V justify potential increases in resource consumption and deployment difficulty? It is recommended that the authors add further discussion on algorithmic efficiency and scalability to provide a more comprehensive perspective on the practicality of the proposed method.

3. The evaluation relies primarily on GPT-4o as a judge to measure helpfulness and safety. While this is a common practice, the inherent bias in “LLM-as-a-Judge” approaches may not accurately reflect the full complexity of real human preferences. In the absence of any human evaluation to calibrate or validate the automatic assessment results, the reliability of the paper’s conclusions is somewhat limited. The authors are encouraged to provide supplementary analysis on the consistency between GPT- and human-based evaluation.

4. The overall fluency of the manuscript could benefit from a thorough proofread. For example, layout and alignment issues in Table 3 detract from the clarity of data presentation, and there are some minor typos and grammatical errors (such as “+DPO (Helpful")”). Correcting these will enhance the professionalism of the paper.

In summary, this is an good paper that addresses the critical issue of MLLM safety alignment in a timely manner and provides a thoroughly validated, systematic solution. The contributions are significant and the experiments are sufficient and robust. This framework meet the current needs of vision-language modalities, and paves the way for exciting new directions in future multimodal AI safety research. Extending these principles to more complex modalities such as video and audio will be a key next step for the field. This work lays a solid foundation for tackling these forthcoming challenges.

I recommend acceptance of this paper and believe that, should the authors address the suggestions above, its quality will be further improved.

---

> ### Author Rebuttal · Authors · 2025-07-31
>
> Dear Reviewer Dc2g,
>
> Thank you very much for your insightful comments. We are encouraged by your positive comments regarding our work establishing an important benchmark for the field, the high quality of our dataset as a valuable resource for future research. We hope the following responses will address your concerns.
>
> ---
>
> > **Q1: deeper discussion of certain key issues**
>
> **R1:** Thank you for your constructive suggestions. We have addressed your comments in detail, and all corresponding revisions will be incorporated into the updated manuscript to ensure a more comprehensive and in-depth discussion of the raised concerns.
>
> ---
>
> > **Q2: Insufficient Depth in Analyzing Training Instability**
>
> **R2:** Thank you very much for your valuable suggestions. We provide a quantitative comparison analysis of the cost model between Vicuna-7B (text-only) and LLaVA-1.5-7B (multimodal), both trained under exactly the same conditions using the Safe RLHF pipeline, without introducing the scaling hyperparameter k proposed in this work for multimodal settings.
>
> As shown in the Table 1, cost model in the multimodal setting deteriorates compared to the text-only:
> - 1) across multiple distributional metrics, the separation between safe and unsafe responses clearly decreases: the mean difference between unsafe and safe responses drops from 5.069 to 2.237; ROC-AUC decreases from 0.958 to 0.888; the overlap coefficient increases from 0.211 to 0.371.
> - 2) the decision boundary between safe and unsafe distributions shifts away from zero: the optimal classification threshold moves dramatically from −0.054 to +0.656. These consistent degradation patterns indicate that multimodal inputs lead to severe compression and shift in the cost distribution.
>
> Therefore, due to the distribution compression and decision boundary shift, the traditional cost model and Safe RLHF training framework designed for text-only models can no longer effectively address this instability. To this end, 1) we introduce the scaling hyperparameter k to mitigate optimization difficulties caused by reduced separability; 2) we introduce the budget bound mechanism to adapt to the non-zero optimal classification threshold. Together, they enable stable and reliable constrained optimization in multimodal scenarios.
>
> **Table1.Performance Degradation of the Cost Model in Multimodal vs. Text-Only Settings.**
> |Metric|**Text-Only(Vicuna-7B)**|**Multimodal(LLaVA-1.5-7B)**|
> |----|----|----|
> |**ΔMean(Unsafe-Safe)$\uparrow$**|**5.069**|2.237|
> |**ROC-AUC$\uparrow$**|**0.958**|0.888|
> |**Overlap Coefficient$\downarrow$**|**0.211**|0.371|
> |**KSS tatistic$\uparrow$**|**0.788**|0.623|
> |**Jensen-ShannonDivergence$\uparrow$**|**0.661**|0.433|
> |**Wasserste in Distance$\uparrow$**|**5.069**|2.237|
> |**Optimal Classification Threshold [1]**|**-0.054**|+0.656|
>
>
> [1] Youden, W. J. (1950). Index for rating diagnostic tests. Cancer, 3(1), 32-35.
>
> ---
>
> > **Q3: Lack of Discussion on Training Cost and Scalability**
>
> **R3:** We sincerely thank the reviewer for their valuable comments. The training cost of the Safe RLHF-V framework is indeed an important issue worthy of further investigation. In response, we conducted a detailed comparison between Safe RLHF-V and conventional vision-based RLHF within the Verl framework. Specifically, we used the Qwen2-VL-7B model and trained on the BeaverTails-V dataset (18k samples) using both PPO and Safe RLHF-V for 1 epoch. The training was performed with a batch size of 4, a maximum response length of 512, and rollout_n = 1, on 8 A100 GPUs.
>
> Under the same experimental settings, PPO took 6 hours with a reduced memory footprint of 17,750 MB per GPU. Safe RLHF-V, trained under the same settings, took 7.57 hours, which is 26.1% slower than PPO.
>
> **Table 2. Comparison of Safe RLHF-V and RLHF (in vision) Training Cost on Qwen2-VL-7B with the Beavertails-V (18k) Dataset**
> |Method|Time|#ofGPUs|Time-Compared-to-PPO-Default|Epochs|Batch-Size|Max-Response-Len|
> |---------------|-----|-------|----------------------------|------|----------|----------------|
> |PPO(invision)|6.0h|8|~|1|4|512|
> |SafeRLHF-V|7.57h|8|+26.1%|1|4|512|
>
> ---
>
> > **Q4: the consistency between GPT- and human-based evaluation**
>
> **R4: ** Thank you very much for your insightful review and thoughtful comments. You raised a critical concern: `whether using GPT-4o as an automatic evaluator is sufficiently reliable and truly reflective of human preferences`. **We fully share your concern.** Relying solely on large language models as evaluation proxies does carry potential risks of bias, especially when dealing with complex and nuanced human preferences where LLM judgments may not fully substitute for human assessments.
>
> To address your concern, as well as that of Reviewer NsRm, we conducted an additional large-scale human-AI agreement study during the rebuttal phase. The setup is as follows:
> - We randomly sampled 2,000 preference annotation instances from outputs generated by various models.
> - All samples were independently evaluated by our long-term professional annotation team specializing in safety evaluation. This team has extensive experience in large model safety assessments and has participated in multiple relevant projects, ensuring both professional judgment and ethical awareness.
> - We then compared the human annotations with GPT-4o’s automatic evaluations to assess the alignment between the two.
>
> > **Note:** Due to our long-standing collaboration with this professional team, we were able to conduct this emergency annotation task within a short timeframe.
>
> The results show strong agreement between human annotators and GPT-4o across both overall safety/helpfulness preferences and fine-grained safety levels (Minor / Moderate / Severe). Specifically:
>
> **Table 1: The human and AI annotation agreement in different safety levels.**
> |                         | minor | moderate | severe |
> |-------------------------|-------|----------|--------|
> | Human and AI Agreement | 85.3% | 87.8%    | 89%    |
>
> **Table 2: The human and AI annotation agreement in total safety and helpfulness preference.**
> |                         | safety  | helpfulness  |
> |-------------------------|-------|----------|
> | Human and AI Agreement | 87.4% | 91.2%    |
>
> As you also noted, several recent high-profile studies have adopted LLMs as evaluation agents, including:
> - OpenAI’s GPT-4 Technical Report;
> - Anthropic’s Constitutional AI;
> - XAI's Grok4;
>
> These works demonstrate that, when large-scale human evaluation is prohibitively expensive or time-consuming, using LLMs as evaluation proxies offers a scalable and practical alternative.
>
> ---
>
> > **Q5: some minor typos and grammatical errors.**
>
> **R5:** Thank you for pointing out the issues regarding clarity and formatting. We acknowledge the layout problem in Table 3 and the minor typographical errors (e.g., “+DPO (Helpful”)”), and we will address all of them through a careful proofreading pass to improve the overall fluency and professionalism of the final version.
>
> ---
>
> > **Q6: Ethical Considerations**
>
> **R6:** Thank you for your attention to the ethical aspects of our work. Regarding ethical compliance, this study has been approved by the Institutional Review Board (IRB), ensuring that the entire data collection and annotation process adheres to established ethical standards and relevant legal regulations. In addition, we offered professional psychological counseling services, which remained continuously available throughout the entire project period to help annotators manage emotional stress or psychological burdens that might arise.
>
> ---
>
> > **Q7: Insights from Decoupling Preferences, generalized to other alignment challenges, such as honesty or fairness in AI?**
>
> **R7:** Thank you for your thoughtful engagement with our “preference disentanglement” approach and for recognizing its significance. A key insight we derived from this method is that many alignment challenges are fundamentally not single-objective optimization problems, but rather involve complex trade-offs among competing human values. Structuring these objectives into explicitly controllable dimensions—and modeling their interrelations during training or evaluation—can lead to more robust and controllable model behaviors. We believe this idea has broader applicability. For example, similar tensions exist between honesty and politeness, or fairness and personalization—domains where disentangled modeling could help better coordinate outputs that reflect diverse human values.
>
> ---
>
> > **Q8: Generalization of the Guardrail Model**
>
> **R8:** Thank you for your interest in the generalization ability of Beaver-Guard-V. Beyond the baseline experiments reported in the original paper, we have further introduced two additional challenging benchmarks—JailBreakV-28k [1], VLSBench [2]. These benchmarks are designed to systematically evaluate the model’s ability to detect broad-spectrum safety risks. Experimental results demonstrate that Beaver-Guard-V significantly outperforms state-of-the-art baselines such as LLaMA-Guard-3-11B-Vision across all two benchmarks. The detailed results are summarized in the following table:
>
> **Table.3:BeaverGuard-V in Jailbreaking Benchmark.**
> |Models|Metrics|vls bench|JailBreakV-28k|
> |------------------------|:------:|:------:|:------------:|
> |Llama-Guard-3-11B-Vision|Accuracy|0.76|0.51|
> |Beaver-Guard-V(Ours)|Accuracy|0.78|0.92|
> |Llama-Guard-3-11B-Vision|F1-Score|0.86|0.48|
> |Beaver-Guard-V(Ours)|F1-Score|0.87|0.95|
>
> Reference:
> [1] JailBreakV: A Benchmark for Assessing the Robustness of MultiModal Large Language Models against Jailbreak Attacks
> [2] VLSBench: Unveiling Visual Leakage in Multimodal Safety

---

> ### Comment · Reviewer_Dc2g · 2025-08-04
> **Response from reviewer Dc2g**
>
> Dear Authors:
>
> Thank you for your detailed responses during the rebuttal period, as well as the substantial additional experiments and discussions you provided.
>
> I am particularly impressed by the authors’ ability to conduct human-AI consistency evaluations within the limited rebuttal stage. It is exciting to see that the consistency between human and AI evaluations has exceeded 85%, which is a rare achievement in real-world settings.
>
> I found that both BeaverTails-V and the Safe RLHF-V pipeline are highly effective in enhancing the safety of MLLMs. **This practical impact is a key reason I strongly recommend accepting this paper.**
>
> I would also like to express my appreciation to the authors for conducting an emergency human evaluation consistency experiment during the rebuttal phase and providing annotated preference data on safety and helpfulness from a professional evaluation team. The results show that across different safety severity levels (mild, moderate, severe) and in overall preferences, the average agreement rate between GPT-4o and human annotations is consistently between 85% and 91%, demonstrating the high-quality and reliability of dataset.
>
> These findings directly address my concerns regarding potential bias in using LLMs as evaluators and provide solid empirical support for the paper’s evaluation methodology. I sincerely thank the authors for their prompt response and rigorous experimentation, which clearly reflect their commitment to addressing reviewer feedback.
>
> For the revised version, I hope the authors can consider incorporating the following aspects:
> 1. The analysis of human-AI agreement results;
> 2. The discussion of the generalization capabilities of Beaver-Guard-V;
> 3. A discussion on the instability of multimodal training;
> 4. A further elaboration on the methodological significance of the “helpfulness–safety preference decoupling” strategy. This is especially meaningful in real-world scenarios, where alignment tasks often involve multi-objective optimization with conflicting values (e.g., honesty vs. politeness, fairness vs. personalization). We have recently been exploring the application of the Safe RLHF-V framework in such multi-objective value alignment settings, and I hope the authors can expand on this direction in the Discussion section.
>
> In summary, Safe RLHF-V not only presents a novel framework for multimodal safety alignment, but also releases a dataset and a multi-level guardrail system for community research. **Given the scientific novelty and practical value of this work, I am raising my score to full accept and increasing my confidence. I firmly believe this paper is worthy of acceptance.**

---

> > ### Author Response · Authors · 2025-08-05
> > **Thank you very much, but it seems there has been some misunderstanding**
> >
> > Thank you very much for your recognition and understanding. We also appreciate your kind support sincerely. However, Reviewer yjaW has raised concerns about your public defense of our work and questioned the motivation behind such actions.
> >
> > During the rebuttal period, we strictly adhered to all guidelines and followed NeurIPS submission policies and academic standards throughout the entire process. While we are unsure whether this misunderstanding might impact the overall review process, we sincerely hope you would consider removing the defense and, if possible, help clarify the misunderstanding.
> >
> > We truly understand and appreciate your good intentions, and we are grateful for your recognition. Apologies again for the inconvenience, and we’d be happy to further communicate if you have any additional questions.

---

> > > ### Author Response · Authors · 2025-08-07
> > > **Thank You for Your Mediation**
> > >
> > > We are sincerely grateful for your kind mediation. Reviewer yjaW has recognized the value of our work and acknowledged our explanation. We deeply appreciate the extra time and effort you devoted to helping reach this resolution.

---

### Note · Authors · 2025-08-11

Dear Reviewers, AC, SAC, and PC

We sincerely thank all four reviewers for their participation in the rebuttal process and for providing constructive comments on our work. We also greatly appreciate the AC for engaging closely with the authors throughout this process.

Following an in-depth rebuttal discussion, the reviewers unanimously expressed positive feedback on our work. In the revised version of the paper, we will make the following changes:

1.	Add an analysis of the challenges related to instability during multimodal safety alignment training (Reviewer Dc2g yjaW).
2.	Further analyze the advantages of dynamic updates, incorporate the additional experiments presented in the rebuttal into the revised version, and include comparisons with other safety-constrained RL methods (Reviewer Dc2g yjaW KXX1 NsRm).
3.	Provide a detailed analysis of training cost in the experimental details section, along with an examination of the guard model’s generalization capabilities and a comparison of real-time inference performance (Reviewer Dc2g NsRm).
4.	Expand and refine the analysis of human–AI consistency, covering three aspects: the realism of the data, the consistency of safety labels (minor/moderate/severe), and related ethical considerations (Reviewer Dc2g, yjaW, NsRm; Ethics Reviewer Kgxu,AHzJ).
5.	Discuss the potential of applying Constrained Preference Optimization to honesty and fairness, as well as to other modalities (e.g., speech, video) (Reviewer Dc2g, NsRm; Ethics Reviewer AHzJ).
6.	Strengthen the ethical considerations section and correct typographical errors in the manuscript (Reviewer KXX1).
7.	Include a dataset statement and describe annotator safety protocols (Ethics Reviewer Kgxu).

**This has been a truly rewarding and meaningful rebuttal process**. Over the course of **31 conversations** with the reviewers, we carefully addressed each concern raised, and the reviewers commented that "their concerns have been solved". We are deeply grateful to the SAC, AC, and reviewers for providing us with this platform and for dedicating your valuable time and effort. We genuinely hope you will recognize our work and support it for a better presentation opportunity.

---

**Once again, we sincerely thank you for the time and effort you have dedicated to our work.**

Best Regards,
Paper authors.

---

### Decision · Program_Chairs · 2025-09-17

**Decision:**

Accept (poster)

**Comment:**

The paper proposes Safe RLHF-V, a framework to improve safety alignment in multimodal large language models by jointly optimizing helpfulness and safety. It introduces (1) BeaverTails-V dataset with dual annotations for helpfulness and safety, (2) Beaver-Guard-V guardrails against unsafe queries, and (3) a constrained optimization approach improving both safety and helpfulness.

This paper addresses a key and understudied problem in MLLM safety alignment through a systematic framework and dataset. Despite limitations, the contributions are valuable and timely with empirical backing.

Authors addressed reviewers’ concerns extensively, including additional experiments, clarifications, and ethical commitments. Some reviewer integrity concerns were raised and forwarded to SAC for review.